# The Impact of Preoperative and Postoperative Nutritional Interventions on Treatment Outcomes and Quality of Life in Colorectal Cancer Patients—A Comprehensive Review

**DOI:** 10.3390/medicina60101587

**Published:** 2024-09-27

**Authors:** Barbara Piekarska, Mateusz Prusisz, Marcin Włodarczyk, Jakub Włodarczyk, Mateusz Porc, Inez Bilińska, Kasper Maryńczak, Łukasz Dziki

**Affiliations:** Department of General and Oncological Surgery, Faculty of Medicine, Medical University of Lodz, 90-213 Lodz, Poland; barbara.piekarska@onet.pl (B.P.); m.j.f.prusisz@gmail.com (M.P.); 007wlodarczyk@gmail.com (J.W.); mateusz.porc@gmail.com (M.P.); inez.bilinska@gmail.com (I.B.); kmarynczak@gmail.com (K.M.); lukasz.dziki@umed.lodz.pl (Ł.D.)

**Keywords:** colorectal cancer, nutritional interventions, postoperative complications, quality of life

## Abstract

Colorectal cancer (CRC) is one of the most prevalent cancers worldwide, with high morbidity and mortality rates. Nutritional status has emerged as a significant factor influencing the prognosis and survival of CRC patients. This comprehensive literature review examines the role of nutritional support in improving treatment outcomes, including the efficacy of interventions, patient quality of life (QoL), and the modulation of inflammatory responses. The findings suggest that tailored nutritional interventions improve clinical outcomes, enhance QoL, and reduce treatment-related complications, particularly by attenuating inflammation. Furthermore, the review highlights the cost-effectiveness of nutritional strategies and identifies key methods to enhance patient compliance with dietary recommendations. In conclusion, integrating nutritional support into CRC treatment plans is crucial for optimizing clinical management and improving patient well-being.

## 1. Introduction

Colorectal cancer (CRC) is one of the most prevalent forms of cancer worldwide, with significant morbidity and mortality rates [1]. The prognosis and survival rates for CRC patients are influenced by various factors, including the stage at diagnosis, treatment modalities, and patient-related factors such as age, comorbidities, and nutritional status. Among these, nutritional status has emerged as a critical determinant of treatment outcomes, influencing both the efficacy of therapeutic interventions and the overall quality of life (QoL) of patients. Nutritional interventions before and after surgery, as well as during other treatment modalities like chemotherapy, play a pivotal role in enhancing the body’s resilience to the stress of cancer treatments, supporting immune function, and promoting recovery. Malnutrition, which is common among CRC patients due to the disease itself and its treatments, can lead to poorer outcomes, including increased postoperative complications, prolonged hospital stays, and reduced survival rates. Therefore, understanding and optimizing nutritional interventions is essential in the holistic management of CRC patients. This paper aims to explore the impact of preoperative and postoperative nutritional interventions on the treatment outcomes and QoL in CRC patients. It will review the current literature and clinical trial data to identify effective dietary strategies and the role of specific nutrients and supplements in the treatment and recovery process. This paper will also consider how personalized nutrition plans, tailored to individual genetic, metabolic, and microbiome profiles, can enhance treatment efficacy and patient well-being. By examining the intersection of nutrition and medical treatment, this study seeks to provide comprehensive insights into how dietary interventions can be integrated into standard CRC treatment protocols. The ultimate goal is to improve clinical outcomes and support the overall health and QoL of CRC patients through evidence-based nutritional strategies.

## 2. Methodology

This comprehensive review was conducted with the objective of synthesizing existing evidence on the impact of preoperative and postoperative nutritional interventions on treatment outcomes and QoL in CRC patients. The methodology followed the Preferred Reporting Items for Systematic Reviews and Meta-Analyses (PRISMA) guidelines to ensure thoroughness and transparency but also included case reports, retrospective studies, RCT, and reviews. The literature search strategy involved a comprehensive search of multiple electronic databases, including PubMed, Cochrane Library, Web of Science, and Scopus. The search used a combination of keywords and Medical Subject Headings (MeSH) terms such as “colorectal cancer”, “nutritional interventions”, “diet therapy”, “preoperative nutrition”, “postoperative nutrition”, “dietary supplements”, “treatment outcomes”, and “quality of life”. These terms were tailored to each database to ensure a thorough search. Inclusion criteria for the review were studies published in peer-reviewed journals, written in English, and published within the last 15 years. The included studies involved adult CRC patients and focused on nutritional interventions related to treatment outcomes or QoL. Both randomized controlled trials and observational studies were included. Studies focusing on pediatric patients, non-colorectal cancers, or those lacking specific data on nutritional interventions or relevant outcomes were excluded from further analysis. Additionally, case reports, editorials, and commentaries were not considered. The study selection process involved a two-step screening. Initially, titles and abstracts of the retrieved articles were screened for relevance. Subsequently, the full texts of potentially relevant articles were assessed for eligibility based on the predefined inclusion and exclusion criteria. This screening was conducted independently by two reviewers/authors. Any discrepancies between the reviewers were resolved through discussion or consultation with a third reviewer. Data extraction was carried out using a standardized data extraction form to ensure consistency and comprehensiveness. Extracted data included study characteristics (such as authors, publication year, country, and study design), patient characteristics (including sample size, age, sex, and CRC stage), details of the nutritional intervention (type, duration, and timing—preoperative or postoperative), and the outcomes measured (treatment outcomes such as surgical complications, length of hospital stay, overall survival, QoL assessments, immune function, and nutritional status). Key findings and conclusions from each study were also extracted. To assess the quality of the included studies, appropriate tools were used. Each study was independently evaluated by two reviewers, with disagreements resolved by consensus or by involving a third reviewer. For data synthesis, a narrative synthesis was provided to summarize the findings from the included studies, detailing the study populations, types of nutritional interventions, and reported outcomes. As this study involved a review of existing literature, it did not require ethical approval or informed consent from patients. Nevertheless, ethical standards were maintained by ensuring proper citation and acknowledgment of all original sources. By following this rigorous methodology, the review aimed to provide comprehensive and reliable evidence on the role of nutritional interventions in enhancing treatment outcomes and QoL for CRC patients.

The first search was performed on 10 June 2024, and the search was updated on 3 August 2024, with a final revision on 15 August 2024. In total, 50 original papers, 11 meta-analyses, 13 review papers and guideline papers, etc. were used for the final analysis (Figure 1). Each article was reviewed independently by five researchers (B.P., M.P., I.B., M.P., J.W.) for inclusion according to the inclusion and exclusion criteria that follow. Disagreements regarding article selection were resolved through discussion until consensus was reached or resolved by discussion between authors and project coordinators (K.M., M.W., Ł.D.).

## 3. Results

The studies reviewed offer valuable insights into the impact of ONS on patients undergoing CRC surgery, assessing various outcomes such as readmission rates, chemotherapy tolerance, QoL, nutritional status, and postoperative complications.

### Adherence to Dietary Guidelines and Nutritional Compliance

Adherence to dietary guidelines among CRC patients is a multifaceted issue influenced by a variety of factors. Su et al. (2024) and Hubbard et al. (2012) conducted a cross-sectional study on compliance with oral nutritional supplements among postoperative patients with digestive tract tumors. Those studies identified factors influencing compliance, such as educational level, age, educational level, adverse reactions to ONS, and others, emphasizing the importance of tailored nutritional interventions [2,3]. Leach et al. (2024) conducted a cross-sectional study in rural Australia, revealing significant barriers to dietary compliance among cancer survivors. These barriers include limited access to nutritious food, a lack of awareness regarding dietary guidelines, and socioeconomic challenges [Figure 2]. These findings underscore the necessity of targeted interventions that address these specific impediments. Community-based nutritional education and the development of local support systems are essential strategies to enhance dietary adherence among rural cancer survivors [4]. Van Blarigan et al. (2020) explored the feasibility and acceptability of a web-based dietary intervention combined with text messaging for CRC patients. This pilot trial demonstrated that digital interventions could be a promising tool to support dietary adherence by providing continuous guidance and motivation. Additionally, Lazar et al. (2023) highlighted the effectiveness of web-based nutrition resources, emphasizing their role in delivering accessible dietary information and support to cancer patients. These digital platforms can bridge the gap in nutritional education, particularly for those who may face geographical or mobility limitations [5,6]. Moreover, Gong et al. (2023) conducted a systematic review and meta-analysis of mHealth diet interventions in cancer survivors, finding that mobile health (mHealth) interventions significantly improve dietary compliance and overall nutritional status. The integration of technology in dietary interventions offers a scalable solution to promote adherence to nutritional guidelines, making it easier for survivors to manage their dietary needs effectively [7]. QoL is another critical aspect influenced by nutritional compliance. Wang et al. (2023) reported improved QoL among CRC survivors participating in a web-based dietary intervention. The study emphasized that better nutritional adherence not only enhances physical health but also contributes to emotional well-being and overall life satisfaction. Ensuring that cancer survivors receive appropriate dietary guidance and support can therefore have a profound impact on their recovery and QoL [8]. The benefits of ONS in post-discharge patients at nutritional risk have been demonstrated by multiple studies. Tan et al. (2021) found that ONS significantly improve nutritional status and reduce postoperative complications in CRC patients. Similarly, Yang et al. (2020) and Lee et al. (2023) showed that preoperative and postoperative nutritional support, including ONS and immunonutrition, positively impacts clinical outcomes and enhances recovery. These findings advocate for the inclusion of ONS as a standard component of nutritional care plans for cancer patients, particularly those undergoing surgery [9,10,11]. Improving adherence to dietary guidelines and nutritional compliance among cancer survivors requires a multifaceted approach. Combining community-based interventions with innovative digital tools and ensuring access to ONS can significantly enhance dietary adherence and improve the overall health for cancer survivors. Despite the comprehensive approach, the authors of an analysis of “Preoperative Nutritional Support for Patients Undergoing Elective Colorectal Cancer Surgery—Does It Really Work?” by Tesar et al. (2023) found that the preoperative nutritional support did not significantly alter the primary outcomes [12].

## 4. Efficacy of Oral Nutritional Supplements in Postoperative Recovery

The role of ONS in enhancing postoperative recovery among CRC patients has been increasingly recognized as vital in improving clinical outcomes and overall health status [13]. Numerous studies have discussed evidence supporting the efficacy of ONS in various aspects of postoperative care [14,15]. Tan et al. (2020) conducted a randomized clinical trial that underscored the significant impact of ONS on post-discharge patients at nutritional risk following CRC surgery. The study found that patients receiving ONS had better nutritional status and experienced fewer complications compared to those who did not receive supplementation. This aligns with the findings of Rinninella et al. (2020), who, through a systematic review and meta-analysis, demonstrated that nutritional interventions markedly improve the nutritional status of gastric cancer patients, suggesting a broader applicability of ONS across different types of gastrointestinal cancers [9,16]. Moreover, the benefits of preoperative and perioperative nutritional support have been highlighted by several studies. For instance, Lee et al. (2023) showed that preoperative immunonutrition significantly improved postoperative outcomes in colon cancer surgery patients [11]. Similarly, Sorensen et al. (2014) and Sorensen et al. (2020) provided evidence that perioperative omega-3 fatty acid supplements can enhance long-term recovery and reduce postoperative complications, emphasizing the importance of incorporating such supplements into standard preoperative care protocols [17,18]. The effectiveness of specific types of supplements, such as branched-chain amino acids (BCAAs) and probiotics, has also been explored. Cogo et al. (2021) found that supplemental BCAAs are beneficial during the oncological peri-operative period, helping to mitigate muscle wasting and improve recovery. Meanwhile, Chen et al. (2024), Chen et al. (2022), and Araújo et al. (2022) demonstrated the efficacy of oral probiotic supplementation in reducing postoperative surgical site infections, which is crucial for enhancing recovery and reducing hospital stays [19,20,21,22,23]. Burden et al. (2017) compared the effects of preoperative oral nutritional supplementation with dietary advice versus dietary advice alone in weight-losing CRC patients. Their single-blind, randomized controlled trial found that the supplementation group had better postoperative recovery, indicating that oral nutritional supplements can provide substantial benefits when combined with dietary counseling [24]. Furthermore, Gao et al. (2023) conducted a secondary analysis of two randomized clinical trials, revealing that early achievement of energy targets through different nutritional support strategies can significantly reduce nosocomial infections in patients undergoing major abdominal surgery. This highlights the critical role of timely and adequate nutritional support in preventing complications and promoting faster recovery [25]. The economic implications of nutritional support were examined by Maňásek et al. (2016), who reported that high-protein nutritional support not only improved clinical outcomes but also reduced treatment costs for CRC patients [26,27]. This suggests that effective nutritional interventions can provide significant cost savings for healthcare systems, further supporting their implementation in routine clinical practice. In conclusion, the evidence overwhelmingly supports the efficacy of oral nutritional supplements in improving postoperative recovery among CRC patients. These supplements not only enhance nutritional status and reduce complications but also contribute to better clinical outcomes and cost savings. Future research should continue to refine these interventions, focusing on personalized nutritional strategies that cater to the specific needs of individual patients to maximize their recovery potential [28].

### Timing and Method of Nutritional Support: Enteral vs. Parenteral Nutrition

Nutritional support is a critical aspect of perioperative care for patients undergoing major surgeries, such as those for cancer. The early achievement of energy targets after major abdominal surgery has proven to be crucial for patients’ quicker postoperative recovery, for example by lowering the risk of infectious complications [29]. Boelens et al. (2014) demonstrated that early postoperative enteral nutrition significantly reduces the duration of postoperative ileus, anastomotic leakage, and hospital stay length [Figure 3]. These findings underscore the benefits of early nutritional intervention in improving postoperative recovery [30]. He et al. (2022) conducted a prospective, single-center, single-blind, randomized controlled trial to examine the effects of preoperative oral nutritional supplements (ONS) on postoperative early enteral feeding intolerance and short-term prognosis in gastric cancer patients. The study found that patients who received ONS showed a significant reduction in early enteral feeding intolerance compared to the control group [Figure 4]. Additionally, these patients exhibited better short-term prognoses, including quicker recovery times and fewer postoperative complications [31]. This suggests that preoperative oral supplementation can enhance postoperative outcomes by preparing the gastrointestinal tract for early enteral feeding. Klek et al. (2022) investigated the effects of immunomodulating versus high-protein oral preoperative supplements in surgical patients across two centers. This prospective, randomized clinical trial revealed that immunomodulating supplements were more effective in reducing postoperative infection rates and improving immune responses compared to high-protein supplements [32]. The study underscores the importance of the type of nutritional supplement used preoperatively, suggesting that immunomodulating formulas may provide superior benefits in enhancing postoperative recovery and reducing complications. Sowerbutts et al. (2024) reviewed preoperative nutrition therapy in gastrointestinal surgery, emphasizing the role of nutritional support in surgical outcomes [33].

The comparison between enteral and parenteral nutrition is crucial in determining the optimal approach for postoperative care. Enteral nutrition (EN), delivered directly into the gastrointestinal tract, is often preferred due to its ability to maintain gut integrity, stimulate gut-associated lymphoid tissue, and reduce the risk of infections. However, in cases where enteral feeding is not feasible, parenteral nutrition (PN), administered intravenously, becomes necessary. The Spanish Society of Intensive Care Medicine and Coronary Units, along with the Spanish Society of Parenteral and Enteral Nutrition (SEMICYUC-SENPE), provided updated guidelines for specialized nutritional and metabolic support in critically ill patients, underscoring the clinical consensus on this topic [34]. Many authors reviewed enteral versus parenteral nutrition and combinations thereof in ICU patients, offering a broader perspective on nutritional support modalities [35,36,37].

Gao et al. (2024) performed a secondary analysis of two randomized clinical trials comparing early enteral nutrition to early supplemental parenteral nutrition in major abdominal surgery patients. This study provided insights into the optimal nutritional strategies for these patients, indicating that there is no difference in the rate of infectious complications depending on the method of meeting energy targets but also suggesting that enteral nutrition may improve patients’ hematological nutritional status [29]. Gao et al. (2022) explored the timing of supplemental parenteral nutrition in patients undergoing abdominal surgery with poor enteral nutrition tolerance. Their findings indicated that early parenteral supplementation could lead to better postoperative outcomes [38]. Ljungqvist et al. (2022) and Zhu et al. (2022) also contributed to this topic by comparing early postoperative supplementary parenteral nutrition and its effects on recovery [39,40]. Assessing body composition may be a useful tool for evaluating patients’ nutritional status [41]. López-Rodríguez-Arias et al. (2021) assessed body composition as an indicator of early peripheral parenteral nutrition therapy in patients undergoing colorectal cancer surgery within an enhanced recovery program. Their study highlighted that the early implementation of peripheral parenteral nutrition (PPN) led to better maintenance of body composition, particularly lean body mass, which is crucial for postoperative recovery. Patients receiving PPN showed fewer instances of malnutrition-related complications and improved overall outcomes compared to those who did not receive early nutritional support [42].

Timing plays a pivotal role in the effectiveness of nutritional support. Preoperative nutritional interventions, as demonstrated by He et al. (2022) and Klek et al. (2022), can significantly influence postoperative outcomes by improving the patient’s nutritional status before surgery [31,32]. Early postoperative nutritional support, particularly within the first 24–48 h, is essential for maintaining metabolic balance and supporting the healing process. López-Rodríguez-Arias et al. (2021) emphasized the benefits of early PPN, particularly in maintaining body composition during the critical postoperative period [42]. This approach aligns with the principles of enhanced recovery after surgery (ERAS) protocols, which advocate for early initiation of nutritional support to optimize recovery and reduce hospital stays. However, not only perioperative nutritional interventions have proven to be significant for patients’ recovery after the surgery. Tan et al. (2020) investigated the impact of ONS on post-discharge patients at nutritional risk following CRC surgery. This randomized clinical trial highlighted the benefits of such supplements in improving nutritional status and recovery. Similarly, Meng et al. (2020) focused on post-discharge ONS combined with dietary advice for gastric cancer patients, demonstrating positive outcomes in nutritional risk management [Figure 5]. Dingemans et al. (2024) studied the effectiveness of high-protein oral nutritional supplements in enabling cancer patients to meet protein intake recommendations during systemic anti-cancer treatment. The randomized controlled parallel-group study confirmed that most patients could achieve adequate protein intake with these supplements. Carli et al. (2019) assessed the effect of multimodal prehabilitation versus postoperative rehabilitation on 30-day postoperative complications in frail patients undergoing CRC resection. Their randomized clinical trial demonstrated the efficacy of prehabilitation in reducing complications. Van Noort et al. (2024) further examined preoperative nutrition therapy in undernourished patients, supporting the importance of nutritional interventions before surgery [9,43,44,45,46,47].

The studies reviewed indicate that both the timing and method of nutritional support are crucial for optimizing postoperative outcomes in cancer surgery patients. Preoperative oral nutritional supplements, particularly those with immunomodulating properties, can reduce complications and improve short-term prognosis [Figure 6]. In the postoperative setting, the early initiation of either enteral or parenteral nutrition is vital. While enteral nutrition is generally preferred, parenteral nutrition serves as an essential alternative when enteral feeding is not possible. Personalized nutritional strategies, considering the patient’s specific needs and surgical context, are recommended to achieve the best outcomes.

## 5. The Influence of Nutrition on the Inflammatory Response

The intersection between nutrition and the inflammatory response in CRC patients is a critical area of investigation, as inflammation significantly impacts patient outcomes. Various studies have underscored the potential of nutritional interventions to modulate inflammation, enhance recovery, and improve overall prognosis. The role of ONS in managing inflammation among CRC patients has been well documented. Ye et al. (2023) [47] conducted a systematic review and network meta-analysis, which revealed that different nutritional supplements, including omega-3 fatty acids and high-protein formulas, can significantly reduce inflammatory markers. The reduction in inflammation is crucial for minimizing postoperative complications and promoting faster recovery. Similarly, Haidari et al. (2019) [48] demonstrated that the supplementation of vitamin D and omega-3 fatty acids during adjuvant chemotherapy improved nutritional status and reduced inflammation in CRC patients [49,50]. This finding is consistent with the work of Sorensen et al. (2014), who reported that perioperative omega-3 fatty acid supplements can reduce inflammatory responses, thereby enhancing long-term recovery outcomes [17].

Several specific nutritional components have been identified as particularly effective in modulating inflammation. For instance, Pappalardo et al. (2014) highlighted the benefits of eicosapentaenoic acid (EPA) in improving body composition and modulating metabolic responses in cancer patients. EPA has been shown to reduce the production of pro-inflammatory cytokines, thus mitigating the inflammatory response [51]. Moreover, the role of probiotics in reducing postoperative infections and inflammation has been increasingly recognized. Chen et al. (2023) and Araújo et al. (2022) provided compelling evidence through their systematic reviews and meta-analyses that oral probiotic supplementation can significantly decrease surgical site infections, which are often a consequence of heightened inflammatory responses [20,22]. These findings suggest that probiotics can be a valuable addition to nutritional strategies aimed at controlling inflammation.

Cachexia, a common condition in cancer patients characterized by severe weight loss and muscle wasting, is closely linked to inflammation. Brown et al. (2023) reviewed body weight and composition endpoints in cancer cachexia clinical trials, noting that nutritional interventions are crucial in managing cachexia and associated inflammation. This is supported by the work of Malta and Gonçalves (2023), who proposed a trial to assess the efficacy of grape seed flour supplementation in attenuating weight loss during the perioperative period for CRC patients with cachexia. Their study protocol emphasizes the potential anti-inflammatory effects of such nutritional interventions [52,53].

A holistic approach combining various nutritional strategies seems to offer the most promise. Ziętarska et al. (2017) highlighted that high-protein nutritional support can improve the nutritional status and QoL of precachectic oncologic patients, with or without chemotherapy-related toxicity. This comprehensive support is essential for reducing inflammation and improving patient outcomes [54,55,56]. The importance of personalized nutritional strategies was further emphasized by Bertocchi et al. (2024), who conducted a scoping review on non-pharmacological interventions in cancer cachexia. They found that tailored nutritional interventions, including specific dietary components and supplements, can effectively reduce inflammation and improve patient QoL [57].

Enteral and parenteral nutrition also play significant roles in managing inflammation. Abunnaja et al. (2013) reviewed the state of the art in perioperative nutrition, concluding that timely and appropriate nutritional support can significantly reduce postoperative inflammatory responses [58]. This is echoed by Kaška et al. (2021), who examined the impact of early postoperative parenteral nutrition on the intensity of postoperative inflammatory responses, finding beneficial effects on patient recovery [59,60,61,62].

The evidence strongly supports the influence of nutritional interventions on the inflammatory response in CRC patients. From specific supplements like omega-3 fatty acids and probiotics to comprehensive nutritional support strategies, these interventions can significantly modulate inflammation, reduce postoperative complications, and improve overall patient outcomes [63,64,65,66].

## 6. The Impact of Nutrition on Cost-Effectiveness

The economic burden of CRC is substantial, and optimizing cost-effectiveness in treatment and prevention is a priority for healthcare systems globally [67,68,69,70]. Nutritional interventions represent a promising strategy to enhance the cost-effectiveness of CRC management by potentially reducing treatment-related complications, improving patient outcomes, and decreasing overall healthcare costs [71,72,73]. Nutritional strategies in the prevention of CRC can be highly cost-effective. The World Cancer Research Fund/American Institute for Cancer Research (2018) highlights that diet and physical activity modifications can significantly reduce the incidence of CRC [74,75,76]. Anderson et al. (2013) [77] emphasized that promoting dietary changes and physical activity during cancer screening can act as a teachable moment, encouraging individuals to adopt healthier lifestyles and thereby reducing cancer risk and associated healthcare costs. Orange et al. (2021) [73] performed a systematic review and meta-analysis, which demonstrated that dietary and physical activity interventions among adults attending colorectal and breast cancer screenings led to significant improvements in cancer preventive behaviors. These findings suggest that incorporating nutritional counseling and physical activity programs into cancer screening protocols can be an effective, low-cost strategy to lower CRC incidence.

The role of dietary interventions in reducing the recurrence of colorectal polyps, a precursor to CRC, has been studied extensively [76,78,79]. McKeown-Eyssen et al. (1994) conducted a randomized trial, which showed that a low-fat, high-fiber diet could significantly reduce the recurrence of colorectal polyps. This dietary modification not only improves patient outcomes but also has the potential to reduce the long-term costs associated with CRC treatment by lowering the risk of progression to invasive cancer. Additionally, targeted dietary interventions such as the reduction of red and processed meat intake have been shown to be beneficial [80]. Dowswell et al. (2012) designed an intervention to help individuals with colorectal adenomas reduce their intake of these foods, finding that such dietary changes can be effectively implemented with appropriate guidance, thus potentially reducing CRC risk and related healthcare expenses [81].

Nutritional support during CRC treatment has been shown to improve patient outcomes and reduce complications, which in turn can lower treatment costs. Haidari et al. (2019) demonstrated that vitamin D and omega-3 fatty acid supplementation during chemotherapy not only improved patients’ nutritional status but also reduced inflammation and enhanced overall treatment efficacy [50]. Such improvements can lead to shorter hospital stays and a reduced need for additional interventions, thereby enhancing cost-effectiveness. Perrier et al. (2019) assessed the cost-effectiveness of an exercise and nutritional intervention versus usual care during adjuvant treatment for breast cancer, finding that the combined intervention was more cost-effective due to improved patient outcomes and reduced healthcare utilization. Although this study focused on breast cancer, similar principles likely apply to CRC, where improved nutritional support can mitigate treatment-related side effects and enhance recovery [82,83,84].

Screening programs for CRC, when coupled with nutritional interventions, can significantly enhance cost-effectiveness by preventing the progression of precancerous lesions to invasive cancer. Gini et al. (2020) reviewed the impact of CRC screening on cancer-specific mortality in Europe, emphasizing that early detection and prevention through screening are crucial for reducing treatment costs. Incorporating nutritional counseling into these programs can further enhance their effectiveness and economic benefits [85,86,87].

Nutritional interventions hold significant promise for improving the cost-effectiveness of CRC treatment and prevention [Figure 7]. By reducing the incidence and recurrence of CRC through dietary modifications and enhancing patient outcomes during treatment with nutritional support, healthcare systems can achieve substantial economic savings [88,89,90]. Future research should continue to explore and refine these strategies, ensuring that they are effectively integrated into clinical practice to maximize their economic and therapeutic benefits.

## 7. Limitations and Future Direction

While this comprehensive review highlights the significant potential of nutritional interventions to improve treatment outcomes and QoL in CRC patients, several limitations should be considered. Firstly, the heterogeneity in study designs, intervention types, and outcome measures across the reviewed studies makes it challenging to draw definitive conclusions. Many studies utilized different types and dosages of nutritional supplements, varied durations of intervention, and diverse patient populations, which may contribute to inconsistent results. Secondly, most studies relied on short-term follow-up periods, which may not adequately capture the long-term benefits or potential adverse effects of nutritional interventions. Longer-term studies are needed to assess the sustainability of the improvements in QoL and treatment outcomes. Thirdly, there is a lack of standardization in the assessment of nutritional status and intervention efficacy. The use of different nutritional assessment tools and QoL questionnaires limits the comparability of results across studies. Moreover, many studies did not adequately control for confounding variables such as the stage of cancer, comorbidities, and variations in treatment protocols, which can significantly influence outcomes. Lastly, publication bias may be present, as studies showing positive results are more likely to be published, whereas studies with negative or inconclusive findings may be underreported.

Future research should aim to address these limitations through well-designed, large-scale randomized controlled trials with standardized protocols. These studies should use consistent definitions and measures for nutritional status, intervention types, and outcome assessments to enhance the comparability and reproducibility of results. Long-term follow-up is crucial to determine the sustained impact of nutritional interventions on both survival rates and QoL. Additionally, future studies should explore the mechanistic pathways through which nutritional interventions exert their effects, providing a deeper understanding of their role in CRC treatment. Personalized nutrition, tailored to the individual needs and genetic profiles of patients, represents a promising area for future research. Investigating how genetic variations affect responses to nutritional interventions could lead to more effective, individualized treatment plans. Moreover, integrating nutritional interventions into standard cancer care protocols requires collaboration among oncologists, nutritionists, and other healthcare providers. Developing comprehensive guidelines and training programs can facilitate the implementation of these interventions in clinical practice. Finally, economic evaluations of nutritional interventions are necessary to determine their cost-effectiveness and justify their inclusion in routine clinical care. Understanding the financial implications alongside clinical benefits will be critical for policymaking and resource allocation in healthcare systems. By addressing these future directions, research can provide robust evidence to support the widespread adoption of nutritional interventions, ultimately improving treatment outcomes and QoL for CRC patients.

## 8. Discussion

In CRC patients, nutritional support has proven to be an important factor in enhancing treatment outcomes. Much research demonstrated that implementing ONS in pre- and postoperative periods has a positive influence on reducing the complication rate, length of hospital stays, and risk of malnutrition. The early postoperative achievement of nutritional goals, through various strategies, has proven to be crucial for quicker and better postoperative recovery. Also, the beneficial role of ONS in decreasing inflammatory response cannot be overlooked. High-protein formulas, as well as several nutritional supplements like omega-3 acids, vitamin D, BCAAs, EPA, and probiotics, can significantly reduce the inflammatory response, which is significant while managing treatment-related complications. In CRC patients, dietary interventions have proven to be significant for overall treatment outcomes, patients’ quality of life, and nutritional status. However, as many studies have demonstrated, patients’ nutritional needs are not standardized and often depend on various factors such as age, educational level, nutritional awareness, adverse reactions to ONS, and access to nutritional products. Understanding the meaning of those factors and their influence on patients’ needs may help improve compliance and adherence to dietary guidelines. Numerous studies have shown that web-based nutritional knowledge resources, such as applications or digital platforms, are able to increase the effectiveness of nutritional interventions by ensuring constant access to reliable nutritional information and improving patients’ management of present dietary needs. CRC, due to its frequency, is a great economic burden worldwide. Nutritional interventions can be a promising strategy for lowering the costs of CRC treatment by influencing the length of hospital stays and treatment-related complication rates and reducing the need for additional interventions. An unbalanced diet is a prominent risk factor for colorectal polyps development. Various studies have shown that red and processed meat, as well as a low-fat, high-fiber diet, can lower the recurrence rate of colorectal polyps, as well as their progression to invasive cancer. That implies that lifestyle changes in the population can lower the risk of CRC and thereby also lower the healthcare costs of CRC management.

## 9. Conclusions

This comprehensive review highlights the critical role of nutritional interventions in improving treatment outcomes and quality of life for colorectal cancer (CRC) patients. Adequate preoperative and postoperative nutritional support can mitigate treatment-related complications, reduce infection rates, and expedite recovery. Preoperative nutrition enhances immune function and resilience to surgical stress, leading to better surgical outcomes and shorter hospital stays. Postoperative support helps maintain muscle mass, prevents malnutrition, and alleviates symptoms like fatigue and loss of appetite. The evidence reviewed demonstrates several strengths, such as consistent findings across various studies indicating that nutritional interventions improve clinical outcomes and quality of life for CRC patients. Many studies also emphasize the importance of timing (pre- and postoperative) and the specific nutritional components (e.g., high-protein, immune-enhancing diets). However, the evidence also has limitations, including heterogeneous study designs, varying sample sizes, and inconsistent intervention protocols, which make it difficult to generalize findings. Additionally, many studies have short follow-up periods, limiting the ability to assess the long-term benefits of nutritional support in CRC patients. Regarding the review process, the strength lies in its comprehensive scope, covering a wide range of databases and including studies that assess multiple aspects of nutritional support. However, limitations in the review process should also be acknowledged. We did not record a detailed archive of the search strategy or document how search terms were adapted for each database, which might have affected the reproducibility of the search process. Additionally, the inclusion criteria for studies were broad, which may have contributed to the variability in the quality of evidence assessed. Despite these limitations, we believe the review provides valuable insights into the potential benefits of nutritional interventions for CRC patients. Future research should focus on addressing these limitations by conducting well-designed, long-term studies with more consistent methodologies and longer follow-up periods. Personalized nutrition approaches, based on genetic and metabolic profiles, should also be explored to maximize therapeutic benefits.

The integration of nutritional interventions into standard oncological care will require greater collaboration among healthcare professionals and the development of comprehensive, evidence-based guidelines. In conclusion, while this review demonstrates that prioritizing nutritional support is essential for enhancing treatment outcomes and long-term health in CRC patients, further research is needed to refine these interventions and address the existing gaps in the literature. Future studies should emphasize personalized nutrition plans that cater to the individual needs of CRC patients to maximize therapeutic benefits.

## Figures and Tables

**Figure 1 medicina-60-01587-f001:**
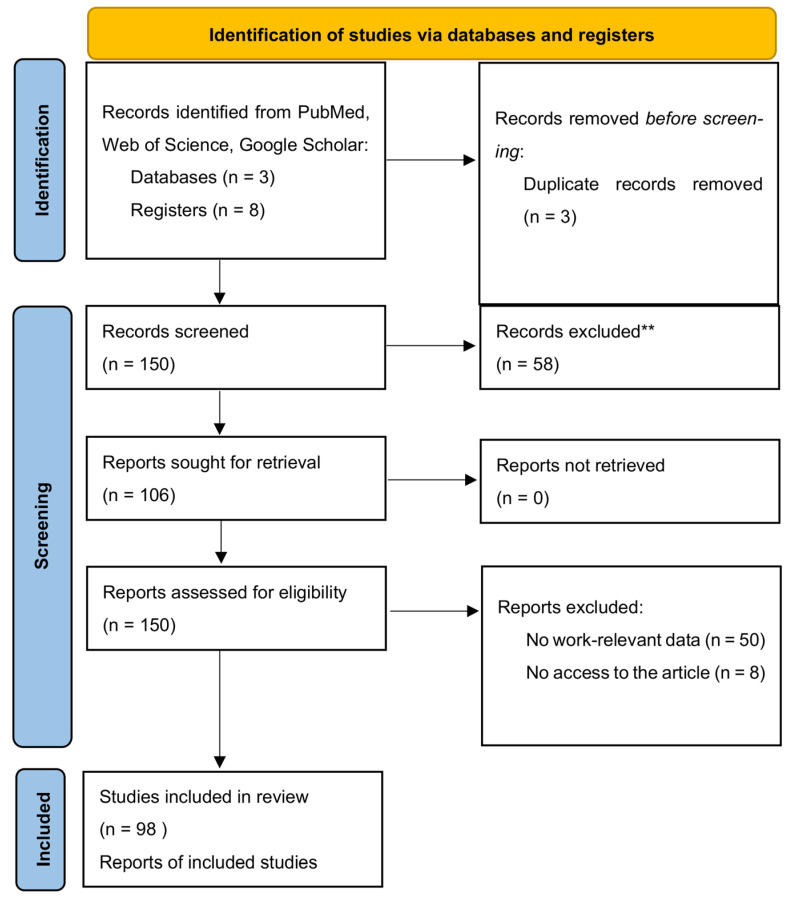
Summary of the evidence search and selection process (flowchart). ** Studies focusing on pediatric patients, non-colorectal cancers, or those lacking specific data on nutritional interventions or relevant outcomes were excluded.

**Figure 2 medicina-60-01587-f002:**
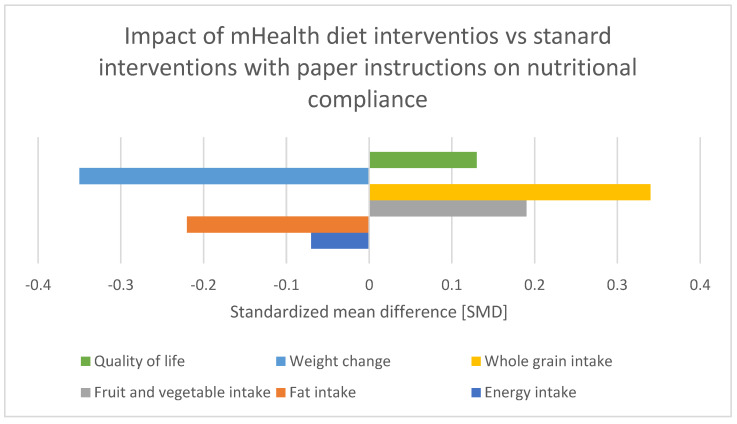
Effect of various mobile health diet interventions vs. standard diet intervention with paper instructions on patient nutritional compliance Standardized mean difference [SMD]. Adapted with permission from Ref. [7].

**Figure 3 medicina-60-01587-f003:**
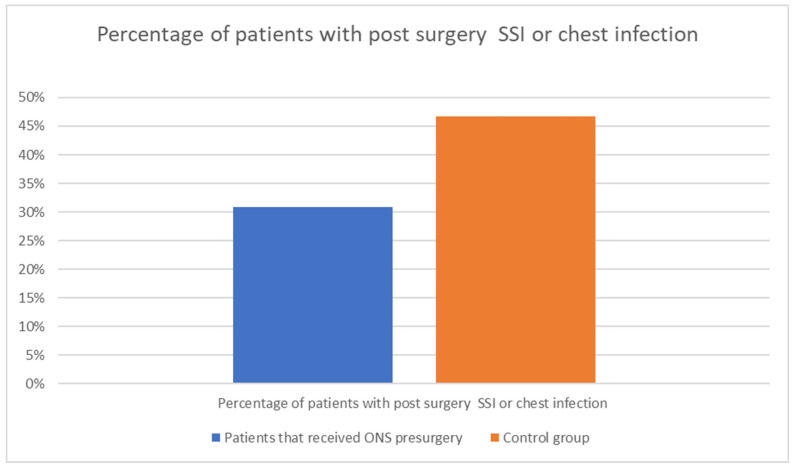
The influence of ONS intake before surgery on the postsurgical complications. Adapted with permission from Ref. [24].

**Figure 4 medicina-60-01587-f004:**
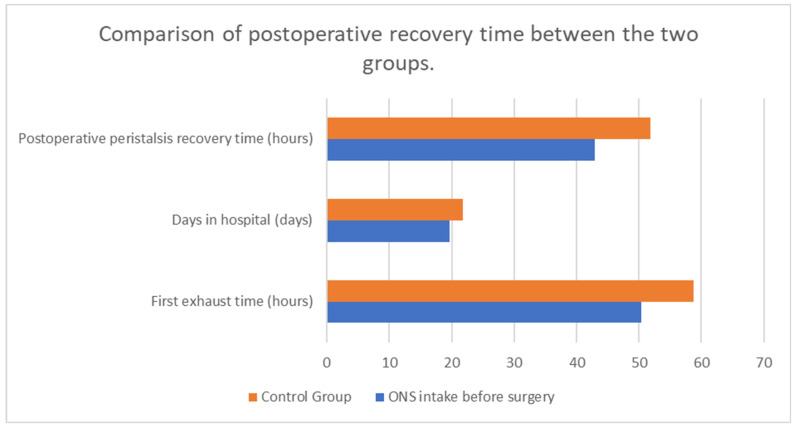
The influence of ONS intake before surgery on postsurgical recovery. Adapted with permission from Ref. [14].

**Figure 5 medicina-60-01587-f005:**
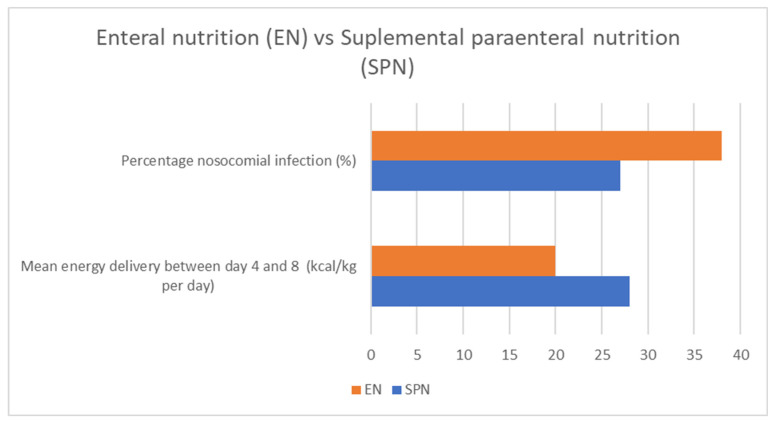
Enteral vs. parenteral food intake and its impact on nosocomial infections and mean energy intake. Adapted with permission from Ref. [35].

**Figure 6 medicina-60-01587-f006:**
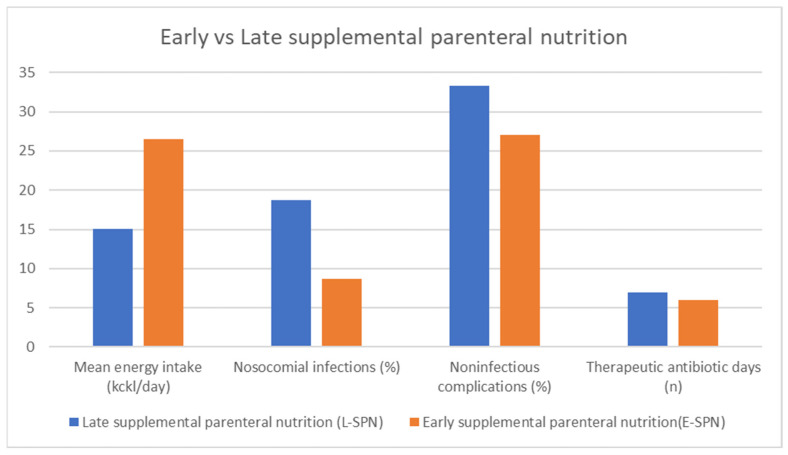
Benefits of early initiation of parenteral nutrition among CRC patients after surgical interventions. Adapted with permission from Ref. [38].

**Figure 7 medicina-60-01587-f007:**
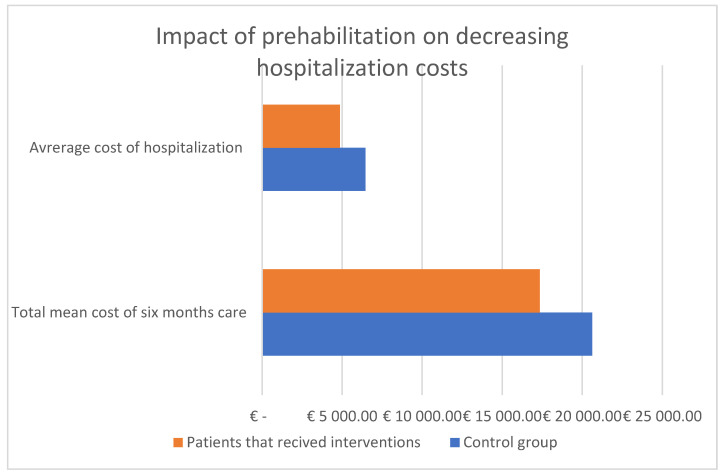
Cost-effectiveness of treatment of patients associated with prehabilitation. Adapted with permission from Refs. [26,84].

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
