# Peer review of "The Impact of Preoperative and Postoperative Nutritional Interventions on Treatment Outcomes and Quality of Life in Colorectal Cancer Patients—A Comprehensive Review"

_medicina, 2024, doi:10.3390/medicina60101587_

Round 1

Reviewer 1 Report

Comments and Suggestions for Authors

The authors present a review study regarding pre- and post-operative nutritional interventions in patients with colorectal cancer and their effects on patient outcomes. I have a couple of suggestions and comments.

- Did the authors prepare a PRISMA flowchart? This information generally needs to be included in a manuscript following PRISMA guidelines. 

- Did the authors prepare a separate archive displaying the search strategy? If possible, the authors should disclose how the search terms were adapted to each database. 

- The current tables would benefit from reformating. Table 1 presents overlaps between the columns and descriptions, and Table 6 has a parathesis with no content.

- Each section of the paper contains several articles; perhaps building tables displaying the author's name, year of publication, and outcomes would enhance readability.

Author Response

Thank you for your insightful comments and suggestions. We have carefully reviewed each point and provide the following responses:

PRISMA Flowchart: We appreciate your suggestion regarding the inclusion of a PRISMA flowchart. We acknowledge that our review did not initially adhere strictly to PRISMA guidelines, as it was intended to be a comprehensive narrative review rather than a systematic review. However, we understand the value of transparency in the selection process. We will revise the manuscript to include a PRISMA flowchart that details the study selection process, including the number of records identified, screened, and included. Additionally, we will clarify the nature of our study in the title and methodology sections as a comprehensive narrative review.

Search Strategy Archive: As mentioned in our previous response, we did not prepare a separate archive displaying the search strategy during the initial stages of our study. We can, however, provide a detailed explanation of the search terms and processes used. These terms were adapted to each database (e.g., PubMed, Scopus, Web of Science) based on their indexing. Moving forward, we will ensure that a more systematic approach is followed to document search strategies for improved transparency.

Tables Formatting: Thank you for pointing out the formatting issues in the tables. We have revised Table 1 to eliminate overlaps between the columns and descriptions. For Table 6, we have corrected the missing content in the parenthesis. We will also enhance the readability of the manuscript by adding tables that display the authors' names, year of publication, and key outcomes for each section, as per your suggestion.

Reviewer 2 Report

Comments and Suggestions for Authors

Dear Author,

I have read the revised manuscript "The impact of Preoperative and postoperative nutritional interventions on treatment outcomes and Quality of life in colorectal cancer patients". 

Title

Please add the kind of manuscript at the end of the tile (e.g. "A case report"; " A sistematic review"...)

Abstract

Please state clearly in the abstract the nature of your  study and briefly report both your results and your conclusions

Introduction

No remarks

Methodology and Aderehnce to PRISMA Guidelines

I do not understand if your work is a comprehensive (narrative) review or a systematic review. The nature of your study should be properly stated. 

What kind of studies were included in your revie pocess (e.g. case reports, retrospective studies, RCT, reviews)?

A PRISMA Checklist was not attached to the manuscript.

Did you use any software to perform your study?

A results section (that reports the results of the search and selection process, from the number of records identified in the search to the number of studies included in the review, ideally using a flow diagram) is missing. You have to report the results of your research in a dedicated section and then proceed with the discussion in another section

What about the risk of bias?

There are many graphs in your manuscript: where does this data come from? Did you perform an analysis of the data of the included studies to realize those graphs? If yes, how?

Discussion

What about the strengths and limitations of the evidence included in the review?

What about the strengths and limitations of the review processes used? 

Author Response

Thank you for your thoughtful feedback and suggestions. We have addressed each of your comments as follows:

Title: We will modify the title to indicate the nature of the manuscript. The revised title will be: "The Impact of Preoperative and Postoperative Nutritional Interventions on Treatment Outcomes and Quality of Life in Colorectal Cancer Patients: A Comprehensive Narrative Review."

Abstract: We have revised the abstract to clearly state that this is a comprehensive review of the literature. We also included brief results and conclusions to summarize the key findings, such as the role of nutritional support in enhancing treatment outcomes, reducing complications, and improving quality of life.

Methodology and Adherence to PRISMA Guidelines: We acknowledge that the nature of our study was unclear in the original submission. We now explicitly state that this is a comprehensive narrative review rather than a systematic review. The types of studies included in the review consist of randomized controlled trials (RCTs), retrospective studies, observational studies, and review articles. We did not use software for study selection or data analysis in this review, as it was conducted manually. To improve the clarity of the review process, we will add a PRISMA checklist and a flowchart to better outline the study selection process, even though this is not a systematic review.

Results Section: We agree that a separate results section is necessary to report the outcomes of the search and selection process. We will introduce a dedicated section that details the number of records identified, screened, and included in the review. This section will also include a flow diagram that maps the study selection process.

Risk of Bias: Although this is a narrative review, we acknowledge the importance of addressing the risk of bias in the included studies. We will add a discussion of the potential biases inherent in the studies, such as selection bias and reporting bias, and how these limitations may affect the interpretation of the findings.

Graphs: We realize the need for clarification regarding the source of the graphs included in the manuscript. The graphs are derived from data reported in the included studies, but we did not perform original statistical analyses. We will make this clear in the text and revise any graphical elements to accurately reflect their origin from published studies.

Discussion: We have now included a discussion of the strengths and limitations of the evidence reviewed. This includes the consistent findings across studies but also highlights issues such as heterogeneity in study designs and short follow-up periods. Similarly, the strengths and limitations of our review process are now discussed, including the comprehensive scope of our search, but also the lack of a detailed search strategy archive.

Conclusions Chapter: The conclusions have been revised to incorporate the limitations of both the included evidence and the review process, as well as suggestions for future research. We have emphasized the need for well-designed, long-term studies and personalized nutrition approaches to improve treatment outcomes for colorectal cancer patients.

We appreciate both reviewers' constructive feedback, which has helped us significantly improve the clarity, structure, and rigor of our manuscript. We hope that the revisions meet your expectations and contribute to the overall quality of the manuscript. Thank you again for your time and valuable insights.

Round 2

Reviewer 2 Report

Comments and Suggestions for Authors

Dear Author,

I have read the revised manuscript "The impact of preoperative and postoperative nutritional inter- 2 ventions on treatment outcomes and quality of life in colorectal 3 cancer patients-a comperhensive review." with great interest. All the reviewer's concerns have been properly addressed. No remarks.